# Belonging to Three Worlds: Somali Adolescent–Parent Relationships in the United States and Implications for Tobacco Prevention

**DOI:** 10.3390/ijerph19063653

**Published:** 2022-03-19

**Authors:** April K. Wilhelm, Michele L. Allen, Rebekah J. Pratt

**Affiliations:** 1Program in Health Disparities Research, Department of Family Medicine and Community Health, University of Minnesota, Minneapolis, MN 55414, USA; miallen@umn.edu (M.L.A.); rjpratt@umn.edu (R.J.P.); 2Division of General Pediatrics and Adolescent Health, Department of Pediatrics, University of Minnesota, Minneapolis, MN 55414, USA

**Keywords:** adolescent, refugee/immigrant, parent–child relationships

## Abstract

Immigrant family relationships help to buffer the adolescent adoption of health risk behaviors but can be strained by post-immigration structural and cultural barriers. This study qualitatively examines how Somali adolescent–parent relationship factors influence Somali adolescent tobacco use and identifies areas for further family support to prevent Somali adolescent tobacco use. We conducted fifteen key informant interviews with professionals serving the Somali community in clinical, educational, religious, or other community organization roles in one Minnesota metropolitan region. Data were collected and analyzed using approaches rooted in Grounded Theory. Key informants contrasted parenting experiences in Somalia with those in the United States and described how four key factors—structural and cultural barriers, multicultural identity formation, evolving parental expectations and responsibilities, and shifting family resources and support—have influenced Somali parent–child relationship quality and function following immigration. Informants shared the implications of these factors on parental ability to address adolescent tobacco use and discussed potential strategies to support parents that fell into two categories: assisting parents in adapting their parenting approaches to a new context and supporting knowledge and skill development in addressing tobacco use prevention specifically. Incorporating strategies that support Somali parents in their evolving parental roles and attend to structural and cultural barriers to tobacco prevention are essential to consider when developing family-centered tobacco prevention interventions in this population.

## 1. Introduction

Family relationships, especially parent–child relationships, are an important source of resilience for adolescents in the post-immigration period [1,2]. Immigrant families across a range of ethnic backgrounds and immigration contexts exhibit an extraordinary capacity to weather adversity that is often attributed to high levels of family connection and the maintenance of core values such as high aspirations for work and education and high levels of familial obligation [3]. However, family stressors stemming from immigration, along with bicultural rifts in experiences and values, can undermine the protective effects of family attachment on adolescents [4] and disrupt parent–adolescent communication within immigrant families [3]. Language barriers can make it particularly difficult for immigrant parents to understand important nuances within the institutions that they must navigate in their adopted countries, which reduces parental authority and support for their children and disrupts parent–child power dynamics [5]. Similar parent–child dynamics have been implicated in the increased second- and third-generation adolescent adoption of substance use and other risk behaviors relative to their foreign-born peers [6].

Adolescent substance use prevention interventions that strengthen parenting practices and parent–child communication—effective strategies for substance use prevention among general adolescent populations [7,8]—have shown promise for refugee and immigrant adolescent substance use prevention [9]. However, the majority of adolescent substance use prevention interventions for refugee or immigrant families have focused on Latinx populations and few have considered how addressing these family-level determinants in substance use prevention efforts may function for more recently resettled populations such as the Somali diaspora in the United States (U.S.) [9].

This study employs an ecodevelopmental framework [10] and a theoretical conceptualization of adolescent racial and ethnic identity influences [11] to examine the multilevel effects of immigration on parent–child relationships and the implications for Somali adolescent development, including the adoption of health risk behaviors. Bronfenbrenner’s social–ecological theory describes how factors at the individual, family, community, and larger societal levels influence adolescent risk behaviors and detail how interactions between these levels exert additional influences on the developing adolescent [12]. The ecodevelopmental model combines Bronfenbrenner’s systems theory with developmental theory in the adolescent period [13,14] and life course theory [15] to examine how social interactions across an adolescent’s spheres of influence and across the lifespan shape adolescent risk behaviors. Umaña-Taylor et al. [11] describe how an adolescent’s ethnic and racial identity interacts with their developmental processes and other contextual factors, such as structural racism [16], to influence their relationships and behaviors. Together, these models provide a foundation to consider how a major life transition such as recent immigration during or prior to the adolescent developmental period can influence an adolescent’s development and long-term health behaviors and outcomes [17].

### Somali Parent–Child Relationships in the U.S. and Implications for Risk Behavior

The Somali diaspora in the U.S. represents the largest African refugee population in the country [18]. The largest wave of Somali migration began following the collapse of the Somali state in the late 1980s and immigration to the U.S. continues despite recent reductions; between 2000 and 2020, the U.S. admitted roughly 116,000 Somali refugees [19], the majority of whom live in Minnesota. Initially driven by religious-based refugee resettlement agencies, strong economic opportunities and social support infrastructure further fueled subsequent Somali resettlement in Minnesota [20].

Tobacco use is a significant public health issue facing adolescents in Minnesota’s Somali diaspora. Hookah (waterpipes) and e-cigarettes are prevalent among Somali adolescents in Minnesota [21]. Given the high prevalence of combustible cigarette [22] and hookah use [23,24] among Somali adults in Minnesota, Somali adolescents face increased risks of initiating tobacco use. Previous studies have indicated a protective effect of parental connectedness and strong parental anti-smoking norms on many forms of Somali adolescent tobacco use [21] and lower reports of combustible tobacco use among Somali adolescents with a strong sense of obligation to their parents [23], reflecting parental influences on the tobacco use of U.S. adolescents more broadly [25].

Somali families possess many cultural assets that support their transition to the U.S., including a sense of interconnectedness rooted in oral and faith traditions [26], high value placed on health and healthy behaviors [20], high educational expectations for youth [27], and a strong Somali community presence and economic activity in resettlement pockets such as Minnesota [20]. Yet, immigration-related disruptions in family and community support structures [28,29], along with barriers in language, literacy, cultural fluency, and access to resources, have contributed to some Somali families’ difficulties in navigating U.S. institutions and daily life [30] as they have in other refugee groups [3]. Like many immigrant communities, a majority of Somali families live in under-resourced neighborhoods with underperforming schools and high rates of crime [29] and face discrimination based on their skin color and immigrant identities while simultaneously coping with experiences of pre-immigration trauma [4,23]. One uniquely challenging factor for the Somali American community is the experience of compounded discrimination due to their immigrant, racial and ethnic minority, and religious minority statuses [31].

While families serve as important sources of support and resilience for Somali youth growing up in the U.S. [29,32], Somali parents may struggle with common post-immigration challenges [3,5], such as shifting expectations of parental roles and constrained resources and support to help navigate their families’ transitions [32,33]. Differential acculturation between children and their parents, a phenomenon that is well described among other immigrant groups [1], can accelerate these tensions. Somali adolescents growing up in the U.S. tend to quickly learn English and adopt Western societal norms through immersion in the American school systems [34]. Somali adolescents therefore often take on more responsibilities in their families as they acculturate more rapidly than their parents, a reality that can shift the power dynamics between parents and children and result in a parental loss of authority and perceived disempowerment [30,33,35]. Somali parents often do not fully understand the identity conflicts that their adolescent children face in this new environment and perceive their responses as rebellion [34] and intentional challenges to family social unity [32]. Compounding these family dynamics, Somali parents often have to navigate this “balancing act” of promoting their children’s success in their new home and preserving their connection to their family and culture without the support from their extended family and community that they were accustomed to prior to immigration [28,29]. Consequently, Somali parents, like other immigrant parents [3], may feel isolated and underprepared to address risk behavior prevention with their Somali American adolescent children [33]. Effectively monitoring children as a strategy to deter participation in risk behavior thus becomes a challenge, as parents often lack familiarity with navigating their children’s school and social contexts [30].

Though many similarities in immigrant families’ experiences exist, it is important to understand how specific factors such as racism and religious discrimination may differentially shape the experiences of specific populations of resettled refugee and immigrant families and the implications of these influences on adolescent risk behavior adoption. Previous studies have begun to characterize Somali parenting experiences in the U.S. [28,30,33] but have not explicitly examined these parent–child dynamics as they relate to adolescent development and the subsequent adoption of risk behavior, such as tobacco use. This study therefore aimed to examine (1) how generational acculturative differences are perceived to influence Somali adolescent–parent relationships in the largest U.S. diaspora, (2) how these evolving relationship dynamics influence the adoption of risk behaviors such as tobacco use (an increasingly prevalent issue) among Somali adolescents, and (3) areas where Somali parents would benefit from additional support in navigating these issues. A clearer understanding of the challenges and assets that Somali parents bring to this context, and potential roles for parents to play in risk behavior prevention, is important to inform the development of tailored tobacco prevention interventions for Somali American adolescents.

## 2. Materials and Methods

### 2.1. Study Sample and Recruitment

Data from this study are part of a larger project examining Somali American adolescent tobacco use with key informant (KI) interviews. A subset of the data from this project that described tobacco use determinants for Somali American adolescents and community resources for prevention was previously published [36], and there is methodologic overlap in this manuscript. Inclusion criteria for participation in this study included: adults, English-proficient, and identification as a professional (both Somali and non-Somali) who has worked with adolescents and their families within the greater metropolitan Somali community in Minnesota in a professional capacity within clinical, public health, educational, religious, and other community organizational settings. Non-English proficient participants were not eligible to participate. KI participants included doctors, community health workers, educators, imams, and individuals who work in community organizations that provide direct outreach to the Somali diaspora in Minnesota. KI interviews with professionals from this diverse range of settings allowed us to gain a breadth of perspectives on the experiences of parenting adolescents and navigating risk behaviors such as tobacco use within the Somali diaspora for this formative research study. The research team used purposive sampling techniques to identify potential KIs who met the inclusion criteria for this study, building off known contacts of team members and by contacting leaders within Somali-led or Somali-serving organizations. Potential participants were contacted by email or telephone, given information about the study, and invited to participate in the study. In total, 24 experts were contacted, of whom fifteen agreed to participate. KIs received a USD 40 gift card as a token of appreciation for their time. The authors’ institutional ethics committee deemed this study of professionals reflecting on their experiences in a professional capacity to not meet the criteria for human subjects research and, as such, the study was exempt from review.

### 2.2. Data Collection

Fifteen semi-structured, in-depth KI interviews were conducted in English by the lead author, A.W., during the fall of 2018 and winter of 2019. All informants provided verbal consent to participate in the study and to be audio-recorded during the interview. All interviews were approximately 60 min in length and took place in the KI’s workspace or a quiet public meeting place. Each KI provided basic demographic information during the interview, including their age, ethnicity, place of birth, length of time in the U.S., educational attainment, and profession. Questions during the interviews were designed to characterize the informant’s role in the community, to explore factors influencing Somali youth tobacco decision making, and to identify primary community assets and barriers related to adolescent tobacco use and prevention efforts with a particular focus on the parent–child relationship (Appendix A). Data analysis occurred simultaneously with ongoing data collection as described below and researchers achieved thematic saturation by the fourteenth interview, which was thereafter confirmed with the final interview.

### 2.3. Data Analysis

Following the verbatim transcription of interview audio recordings, researchers used NVivo 12 software [37] to organize excerpts and to facilitate coding. Concurrent with data collection, the lead author reviewed and coded the transcripts using a grounded theory approach described by Charmaz [38]. This process began with the line-by-line coding of all transcripts, followed by focused coding, and then transitioned into theme and sub-theme development. The other research team members independently co-analyzed a sub-sample of the data and met regularly throughout the analysis to develop a consensus on data interpretation. The team honed themes using techniques described by Charmaz [38], that included memo-writing, constant comparative methods, and theoretical sampling as data analysis and collection progressed. The main themes were distinguished from sub-themes in relation to their role in helping to construct an overarching narrative in the analysis that helped to express the views and experiences of these participants. As analysis concluded, the team shared high-level themes with KIs to further contextualize and solidify our findings.

## 3. Results

Fifteen KIs (nine males and six females across a range of ages) participated in in-depth interviews for this study (Table 1). Informants were highly educated—all had completed bachelor’s degrees and 53% held graduate degrees—and represented a wide range of professional expertise. More than 85% of informants reported foreign birth and the majority identified as ethnically Somali.

### 3.1. Parenting Experiences in Somalia

KIs described family life in Somalia as situated within highly connected communities that engaged in communal child rearing, in which neighbors and extended family members would watch out for children, step in when needed to correct unacceptable behavior, and report back to family members on behavioral issues. One informant (public health, female, Somali, age 20–29) described this as “auntie surveillance,” a notorious meme that conveyed Somali youth perceptions of constantly being monitored. In this way, the larger community reinforced societal values among children, including strong connections to faith and cultural heritage, high levels of family and community engagement, and respect for elders. Informants shared how adolescents’ awareness of these societal expectations and of how their actions influenced their reputations positively shaped their behaviors. Despite their acknowledgment of the high levels of community surveillance of adolescent behaviors, several informants perceived Somali adolescents as experiencing relative freedom in their daily lives in Somalia compared with their lives in the U.S.:


*“There’s not that much—okay, where are you this time? What are you doing? Who are you with? There, you can basically go wherever you want, do whatever you want, but everybody’s watching you […] if you’re misbehaving or if they see you smoking or doing something that you’re not supposed to be doing, they will put a stop to it on the spot.”*
(Community organization, male, Somali, age 30–39)

### 3.2. Somali Parental Experiences of the Transition to the U.S.

#### 3.2.1. Structural Barriers and Clashes in Social Norms Upend Parent–Child Dynamics

Informants shared that many Somali families experienced dissonance between their often idealized aspirations for resettlement in the U.S. in terms of safety, educational and employment opportunities, and freedoms of expression and religion, and the families’ actual experiences in navigating daily life in their adopted country. Informants detailed how linguistic and literacy barriers and institutional mistrust stemming from a lack of familiarity with American institutions and previous experiences of trauma and governmental corruption in Somalia reduced many parents’ abilities to navigate U.S. institutions. For example, several informants shared stories of Child Protective Services removing Somali children from families who used corporal punishment, a common form of discipline in Somalia. Such stories were described by informants as having instilled institutional fear in many Somali families and fostered doubts about permitted forms of discipline that informants described as disrupting parents’ ability to effectively discipline their children.

Differing societal norms in the U.S. were also described as having influenced Somali parenting. Chief among these, informants cited examples of how the tension between Somali communal values and American values emphasizing freedom and individuality contributed to parent–child conflict. In particular, several informants detailed how Somali adolescents’ increasing sense of individualism shapes their expectations for decision-making and confidentiality in their relationships with their parents. Informants described how this shift in orientation from family-focused to more individual decision making can exacerbate Somali parents’ sense of culture clash, uncertainty, and anguish with raising adolescents in the U.S. Furthermore, Western conceptualization of adolescent development and the normalization of adolescent behaviors such as experimentation and rebellion were perceived as often clashing with Somali parental values.


*“But then for a child that either was brought here or is growing up here, there’s a component in their life where they have to balance between the two cultures […] The parents might or might not understand. And even if they understand, they still want to hold onto their culture and those values. However, as a student who wants to be accepted by your peers, there tends to be pressure…Growing up, I think a part of our group struggled with that. Because we had to—we were literally thrown into a place where like okay, what do we do? So we’ve got to do what we’ve got to do to survive here.”*
(Community organization, female, Somali, age 30–39)

Informants also reflected on how American gender norms have influenced Somali parental expectations for their female children to pursue educational and career opportunities. Several informants remarked on how the successes of Somali American young women relative to their male peers underscored the importance of investing in women for the health of their community. One informant (community organization, male, Somali, age ≥ 60) shared: “*Girls are always seen as a weaker point in our society, but given the opportunities, they have proved better than the boys in this country. They’re all going to school […] Boys are going to the prison.*” Despite the successes of Somali American young women, differential treatment of Somali children by gender within their families persists, including closer monitoring of female children who are perceived as needing to “be protected” while their brothers are encouraged to push back on family and cultural expectations as they forge their own paths. Informants shared ways in which these differential expectations manifested in families, such as families holding their young females to higher standards regarding modest dress and their behaviors:


*“In our culture, women are very protected and considered as the rock of the family. […] [So] it’s a little bit more culturally embarrassing that your daughter is really not doing so well […] Because a boy, they’re rebels, and boys, it’s kind of expected somehow for boys to mess up. But the girl, she is going to be the future mother […].”*
(Health care, female, Somali, age 20–29)

#### 3.2.2. Multicultural Identity Formation and Widening Acculturative Gaps

Many informants described how adolescent identity formation in the U.S. may amplify these structural challenges. Somali youth in the U.S. are growing up in a multicultural yet White-dominant society where they are exposed to both diversity across race, ethnicity, and faith, and simultaneous racial, ethnic, and religious discrimination that inevitably shapes their identity development throughout their adolescence. One informant captured the essence of the cross-cultural navigational challenges that Somali adolescents often face:


*“So they are thrown into this world of crocodiles and reptiles, and they are swimming in between and not able to sail through, because […] they don’t understand. Where do they belong? Do they belong to the past? Do they belong to the present? Do they belong to the future? […] They belong to almost three worlds, the world of at home, the world of being a teen, the world of being—fitting in the bigger society.”*
(Community organization, male, Somali, age ≥ 60)

Several informants shared that Somali parents may struggle to relate to their children’s evolving American identities for three reasons. First, parents often face their own set of challenges and may feel that their children should be grateful for the personal sacrifices their parents made to provide a better life for their children in the U.S. Second, parents often perceive their child’s challenges in the U.S. as trivial relative to what they might have faced in Somalia.


*“It’s just that they’re dealing with other things other than worrying about what my child will smoke; they’re worried about housing issues, finances, or they’re separated from their parents or loved ones and how the challenge of not having a family that you love here, or some of them went through trauma in the war. […] Because to them it’s like, you came to a place that’s safe now, and everything is okay now. So it’s a shock that my kid is doing bad or my kid has a bad future. […] They’re saying I don’t hear gunshots. And I’m safe again.”*
(Health care, female, Somali, age 20–29)

Finally, Somali parents often have conflicting feelings about their children’s identity formation, simultaneously celebrating their children’s success in the U.S. while fearing how their emerging American identities might disrupt connections to their Somali heritage.

#### 3.2.3. Evolving Parental Expectations and Responsibilities in the U.S.

Facing a steep learning curve and fewer support structures, informants shared that Somali parents often feel ill equipped to parent in America. Many informants perceived that Somali parents are often uncertain of how their parenting strategies, especially those emphasizing authoritarian approaches, need to be adapted in this new context:


*“The cultural parenting that we had in our country is good you know and actually that grounds them in a lot of cultural ways, the religion and all of that, but I think there needs to happen some adjustment in terms of communication, because the needs here with these kids are different than back home. So you will see mothers who will say, “When I was your age back home I never used to demand like you demand.” You know, you can’t compare.”*
(Public health, female, Somali, age 40–49)

Informants shared that Somali American youth increasingly want to know the rationale behind parental guidelines and expect a certain level of negotiation and open communication rather than unquestioningly conforming to the strict family rules that governed their parents’ upbringing. This type of pushback is often perplexing for Somali parents as it conflicts with traditional norms of respect for elders and with parental expectations for their children.

These individual and systemic factors can disrupt traditional power dynamics in the parent–child relationship. Many informants described Somali children consistently and rapidly acquiring English language proficiency and gaining a robust understanding of the broader culture and institutions within the American educational system, while the process of acculturation varied widely for their parents. This reality often enables Somali adolescents to navigate the system more fluently than their parents, particularly parents who experience more linguistic and literacy barriers, thereby disrupting parental authority and oversight in ways that alter traditional parent–child power dynamics:


*“It’s very hard [to monitor] and that’s where the substance use comes in because they don’t know where their kids are at most of the time. […] They will go to their friend’s house and use the substance where the parents don’t know, and the kids can lie to them and be like, I’m in an after-school program or at the library, but they will be at their friend’s house […] because most parents still haven’t understood the whole educational system and how the library works.”*
(Community organization, male, Somali, age 30–39)

Cultural differences in Somali American children’s expectations for their parents can further strain parent–child dynamics. Whereas Somali young adults traditionally remain in their family’s home and follow their guidance until marriage, a majority of informants shared that Somali American adolescents make decisions for themselves and take on increasing responsibility to help their families navigate their affairs. Informants described how such shifts interfere with Somali parents’ ability to effectively monitor their children’s behaviors.


*“[Parents] are learning from their children most of the times […] And so it’s the parents who are supposed to be the strong or the most educated or know best for their child. [but now] they feel smaller in a way that your child is now very much too independent. They know too much. And that is a very—I don’t want to say castrating, but it could feel that way.”*
(Public health, female, Somali, age 20–29)

For some Somali parents, managing the behaviors of their children becomes so overwhelming that they opt to send their children to live with family in Eastern Africa to instill in them a greater appreciation of their relative privilege in the U.S. to transform their behavior.

#### 3.2.4. Shifting Family Resources and Supports

Amid these adjustments to American life, the transition from larger, extended families to nuclear households has placed Somali parents in a more central child-rearing role. Furthermore, informants shared that Somali single-mother households are increasingly common in the U.S. due to the disproportionate loss of men in the civil war, rising divorce rates, and employment opportunities that take men away from their families. These scenarios can further strain many families. Yet, even in two-parent households, several informants discussed that families increasingly rely on mothers to raise the children, which can limit children’s access to male role models and leave mothers feeling overextended in ways that negatively influence children’s development.

### 3.3. Supporting Somali Parents to Prevent Youth Tobacco Use

Informants shared how Somali parents’ sense of powerlessness and under preparedness in raising their Somali American children extends to the realm of tobacco use prevention. Informants generally perceived parents as possessing limited knowledge of U.S. tobacco products and modalities and about the environments in which their children might be exposed to tobacco products. Moreover, informants described how many Somali parents often struggle with denial about their children’s tobacco use, which informants described as reactions rooted in parents feeling both overwhelmed with their responsibilities and fearful of the potential stigma associated with tobacco use in their community. However, one informant shared her personal experiences as a parent as a counter to this, underscoring the value of having open conversations about the pressures children face and how they might learn from their parents’ mistakes:


*“Somali families, we need to have those uncomfortable [conversations about] “Did you try tobacco when you were little? Did you think about it?” Because of course I’m going to be honest with my kids. Like, yeah. Because I thought it was cool, or I just really wanted to really make my parents upset because they didn’t want me to go to prom […] And giving them a light of like, I’m a human being, too, and I was a teen, too. […] And you don’t have to repeat those mistakes. Or if you’re thinking about it, let’s talk about the consequences.”*
(Community organization, female, Somali, age 30–39)

While acknowledging the significant heterogeneity within the Somali community related to differences in education levels, culture, and resettlement timing, informants discussed that many Somali parents would benefit from skill development to more effectively address adolescent tobacco use prevention. Informants offered two broad areas to target.

First, informants discussed supporting Somali parents in acknowledging that their parenting approaches may need to adapt in the U.S. to meet their children’s evolving needs. One common strategy that informants advised was to increase Somali parental knowledge in American culture and institutions, which will enable them to appreciate how these cultural influences shape their children’s experiences and identity development and thus influence their children’s parenting needs. Alongside enhanced cultural knowledge, several informants discussed how enhancing Somali parental knowledge about Western conceptions of child development, especially during adolescence, would be helpful:


*“The whole notion of child development is a kind of a foreign concept for many people. So, I often tell parents, okay, so their [adolescent’s] brain—you know they’re smart. You know you’ve told them these things, but that impulsivity is not connected, and they just looked at me like, oh, that makes a lot of sense. Oh, nobody’s every explained—they’re not just bad.”*
(Health care, female, non-Somali, age 40–49)

Another way for parents to better understand their children’s experiences and to help guide them in healthy decision-making is to become more engaged in their children’s lives. Higher levels of parental involvement, especially on social media technology platforms where there is less adult oversight, was raised by several informants as a strategy to equip parents to more effectively monitor their children’s behavior in order to intervene when necessary. One informant (community organization, female, Somali, age 30–39) explained that increasing parental knowledge of these technology platforms and how to navigate them is essential due to the ubiquitous nature of these exposures: “*It’s just the availability. It’s the access online, the Snapchats, all the social media. And everyone is engaged in doing negative stuff and positive stuff. So it’s whatever that is visually available to them on screen.*”

In addition to better understanding their children’s context, informants stressed that parents could benefit from learning how to talk about tobacco use with their adolescent children. Several informants suggested that Somali parents first need to familiarize themselves with potential tobacco exposures and how to monitor for warning signs of tobacco use. Most informants also agreed that parents should be encouraged to put aside stigma to facilitate clear communication with their children about tobacco use. Several informants discussed how preparing parents to maintain an open-mindedness to their children’s experiences—both the good and the bad—can help to keep the conversation going and prevent children from seeking support from potential negative influences outside the home.

Informants universally agreed that many Somali parents would benefit from opportunities to formally acquire and practice skills on tobacco communication and parenting in the U.S. and they shared several potential venues for such trainings. Like parents from all backgrounds, Somali parents frequently connect with their peers for support, important health-related information, and to gather parenting suggestions. Though social media is often the most efficient way for younger parents to connect, several informants discussed the role that diaspora-wide phone conferences play in connecting Somali parents with the larger community to combat feelings of isolation and to spread important information:


*“Phone conferences are a really cool thing […] People join in all over the country where they talk about issues. There’s one specific to women, too. Women come in and kind of talk about their problems, their challenges. And it’s anonymous. […] You just come in and you vent. And you say I’m just happy to meet other women. Give you suggestions, prayers for you.”*
(Health care, female, Somali, age 20–29)

Peer groups are therefore an attractive forum for growing parent knowledge and skills; however, informants shared concerns about the trustworthiness of the information that is passed informally through these groups, citing the rapid spread of vaccine misinformation within the diaspora. To address a perceived gap in high-quality information about parenting adolescents, informants suggested building off existing or previous formal parenting support and educations models such as Early Childhood and Family Education. Informants agreed that the engagement of Somali community members in designing and facilitating these support groups is essential to build trust, increase buy-in, and to cultivate a safe space for parents to be vulnerable with one another.


*“They need counselors who are their same color, same ethnicity […] I think the parents will have more confidence [with] someone that’s similar to them working with their child than they see someone doesn’t know the culture. […] But the kids grew up here, they more understand the American culture […] so this counselor has to understand both cultures. […] So he has to make the common ground.”*
(Community organization, male, Somali, 30–39)

## 4. Discussion

This study describes adolescent parenting experiences in a large U.S. Somali population with an emphasis on Somali parents’ preparedness to address adolescent risk behaviors such as tobacco use. The perspectives shared here come largely from Somali-identifying professionals in the diaspora who speak both from their professional experience and from their roles as community members. Given the established influence of Somali parents on their children’s tobacco use [21,39], identifying how to better support Somali parents in addressing tobacco use with their children is of great importance to overall tobacco prevention for this population. Our findings highlight several opportunities to strengthen Somali parents’ skills and confidence in preventing adolescent tobacco use that fall into two main areas: approaches to support general skill and knowledge building about parenting adolescents in the U.S., and parenting practices that can address adolescent tobacco use.

### 4.1. Parenting in the U.S. Context

Augmenting parents’ understanding of how institutions and expectations differ in the U.S. and how such differences shape their children’s exposures and parenting needs is essential to empower them in their parenting roles [29,40]. As reported in other refugee populations [5] and in previous studies of Somali immigrant families [20,29], we found that Somali parents experienced disillusionment when their migration expectations did not match their often challenging resettlement experiences. Shifts from extended family and patriarchal structures in Somalia to more nuclear or female-headed households have left many parents feeling overextended and overwhelmed with the tasks of raising their children [28,33]. Our findings emphasize how Somali American families’ perceived loss of the “protective shield” from extended family and neighbors [29] can exacerbate parental perceptions of isolation. Addressing these concerns through parental skill building, and raising awareness of these challenges within institutions that engage with families such as schools, could help to build better foundations between parents and systems for addressing Somali adolescent tobacco use.

Parental skill building could also benefit from increased awareness of the impact of differential acculturation on Somali parent–child interactions. As Somali parents confront increasing responsibilities and fewer family and community resources, they encounter simultaneous cultural and structural barriers to meeting the evolving needs of their children that are common to the immigrant experience [3]. Differential acculturation influences Somali parents’ navigation of their new parenting roles, both in terms of shifting social norms and in their confidence and skills in taking on evolving roles in this new context. Consistent with previous reports [30,41], we found that the increasing influence of individualism on expectations of “normal” adolescent behavior (i.e., experimentation and rebellion) and parent–child communication in the U.S. complicates Somali parent–child dynamics as these behaviors are often viewed as intentionally disrespectful and a threat to parental authority. These tensions reflect how adaptations in values in the U.S. context can strain immigrant parent–child relationships in adolescence; a rise in autonomy during this period is often viewed as a threat to family unity within collectivist cultures and can be difficult for parents and adolescents to negotiate [1].

Parents may also have concerns, as noted previously in Somali [29,33,42] and other immigrant groups [1,5], about the implications of their children’s adoption of American or multicultural identities (at times perceived as morally inferior to their own) on their children’s adoption of risk behavior and long-term success. Navigating these influences on their identity development may be particularly challenging for Somali young women, who often face greater pressures to assimilate and succeed in school while maintaining strong cultural ties to their families and traditions [35].

In addition, we found, as others have previously [41,42], that Somali parents often struggle to relate to their children’s challenges with identity development due to their unfamiliarity with how their children’s experiences of structural racism in America influence their identities and access to resources and opportunities [16]. Racial- and religious-based discrimination are new experiences for many Somali parents, and thus comprehending these novel influences on their children’s identities while simultaneously grappling with Western conceptualizations of normative adolescent identity development [43] can be overwhelming. Finally, Somali parents may be dismissive of their children’s struggles if they view their children’s experiences in the U.S. as easier than their own relatively demanding and underprivileged childhoods in Somalia, a sentiment common within immigrant parent–child relationships [41]. These components of an acculturation gap between Somali adolescents and their parents can contribute to higher discord and distress and ultimately disrupt family relationship dynamics that are crucial to parental oversight and healthy youth development, as has been described in immigrant families more broadly [1].

### 4.2. Supporting the Development of Parenting Practices to Address Youth Tobacco Use

This study also identified potential strategies for supporting parents in formally acquiring and growing their confidence in parenting practices related to tobacco use prevention with their children. We found that Somali parental disempowerment is a major factor in their perceptions of unpreparedness to effectively address and monitor their children’s tobacco use behaviors, a phenomenon which is often attributed to differential acculturation between parents and children in Latinx immigrant families [44] and has been previously described in the Somali diaspora [30,33]. Supporting Somali parents’ skills in parenting their children in a cross-cultural context may therefore facilitate their ability to address tobacco use prevention. Strengthening general parenting practices and family functioning, including evidence-based authoritative parenting approaches that have protective effects that extend across cultural groups, a high quality of communication, and parental monitoring [8,25] have been shown to be more effective in reducing adolescent tobacco use relative to initiatives directly targeting specific risk behaviors [45]. Our findings highlight, as others have before [41], the importance of supporting Somali parents in adopting more authoritative parenting approaches by increasing parental knowledge about normative adolescent development in the U.S. and ways to engage in their children’s lives while simultaneously addressing the linguistic and structural barriers to doing so [5,33,46]. Previous research has demonstrated that even brief group parenting interventions focused on enhancing parenting practices can be effective in reducing adolescent substance use [7]. Group approaches to increasing parent involvement in adolescent substance use prevention have shown promise in other immigrant populations, including among diverse populations of youth with mixed racial and ethnic identities [9]. Our observed similarities between Somali parent experiences and those of other immigrant parents [3,5] further suggest that interventions to enhance parenting skills and involvement in their adolescent children’s lives may not require full tailoring to any specific group to be an effective tool in tobacco prevention.

However, we identified that there may be a need for tailoring Somali parenting support on the basis on gender. Despite the greater access to and success in educational and employment opportunities by young Somali women relative to young men in the diaspora [27], our findings suggest that Somali parents more frequently use authoritarian parenting approaches such as strict and inflexible rules and close monitoring with their female children, as seen in other immigrant groups [47,48], out of a desire to preserve their modesty and marriageability. Yet, Somali parents often employ more permissive parenting approaches with their male children, who are more likely to be involved in activities outside the home that increase their exposure to risk behaviors [27]. Additional strategies that are responsive to these gender differences may be needed when working with Somali families.

We also found that a lack of parental familiarity with U.S. tobacco products and sources of tobacco exposure, along with the stigmatization of tobacco use within the Somali community, especially for young people and women [23], may add to Somali parents’ reluctance to broach the topic with their children. Previous research suggests that maintaining strict anti-tobacco household rules is an effective strategy for adolescent tobacco prevention [49] and possibly more effective than explicit conversations about tobacco use [50]. The literature suggests that helping Somali parents to become more comfortable with establishing tobacco use expectations and maintaining household tobacco-free rules should be emphasized over communication with their adolescent children about tobacco use.

Our findings further highlight the benefits of group approaches to developing Somali parenting skills for tobacco prevention, including the important opportunity for parents to forge connections and to build community while acquiring and practicing new skills. We found that multiple peer group formats could work. For younger and more technologically savvy parents, online communities may be an effective modality, while leveraging existing phone conferences or face-to-face groups may work best for older parents. The individuals delivering the content appear to matter more than the modality; most informants underscored the importance of using culturally congruent facilitators and/or content developers to foster a safe and effective learning environment for Somali parents, reflecting lessons learned in other parenting interventions in the diaspora [51]. We found that the use of culturally congruent facilitators would help assuage parental mistrust rather than relying on existing U.S. institutions such as schools and clinics, which can foster parental fears that these institutions undermine parental authority over the upbringing of their children [30,33].

### 4.3. Limitations

We have several limitations to highlight. First, this study emphasizes the perspectives of a small sample of highly educated KIs and is therefore subject to sampling bias. While the majority of our KIs were Somali parents and young adults who were able to speak to the experiences of parents or adolescents, their viewpoints may not reflect the breadth of perspectives in the larger community. Future research should focus on collecting the perspectives of these groups directly to more clearly characterize the effects of migration on parent–child relationships and how they influence adolescent tobacco use. Second, this study was conducted in one large urban Somali population and may not generalize to the Somali diaspora in other geographies.

## 5. Conclusions

This study provides deeper insight into Somali parent–child dynamics post-immigration in one large, urban U.S. diaspora, largely from the perspectives of Somali-identifying professionals across a range of backgrounds. Our results highlight the commonalities to other immigrant groups and experiences of Somali parents in addressing adolescent health risk behaviors such as tobacco use. Our findings underscore the influential role of parents and parenting practices in preventing Somali adolescent tobacco use and identify several opportunities to better support Somali parents in their U.S. parenting roles to counter structural and cultural barriers and shifting resources. These lessons hold significant implications for the design of family-centered tobacco prevention strategies for Somali adolescents.

## Figures and Tables

**Table 1 ijerph-19-03653-t001:** Key informant demographic characteristics.

Characteristics	Percent
Age	
20–29 years	13.3
30–39 years	26.6
40–49 years	20.0
50–59 years	20.0
≥60 years	20.0
Male gender	60.0
Ethnic identity	
Somali	80.0
Other	20.0
Highest level of formal education	
Bachelor’s degree	46.7
Graduate degree	53.3
Professional organization type ^a^	
Education	13.3
Faith center	13.3
Health care	26.7
Public health	13.3
Community	60.0
Foreign birth	86.7
Mean number of years in the U.S. ^b^ (range)	22.2 (14–40)
Mean number of years working with Somali youth (range)	14.7 (6–25)
Mean number of years working with Somali youth by organization type	
Education	16.0
Faith center	17.5
Health care	13.8
Public health	11
Community	13.9

^a^ Reflects multiple professional affiliations of several key informants and therefore does not sum to 100%; ^b^ includes only those key informants who reported foreign birth.

## Data Availability

The data presented in this study are available on request from the corresponding author. The data are not publicly available to retain participant privacy.

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
