# Peer review of "Belonging to Three Worlds: Somali Adolescent–Parent Relationships in the United States and Implications for Tobacco Prevention"

_ijerph, 2022, doi:10.3390/ijerph19063653_

Round 1

Reviewer 1 Report

This is a very well-written manuscript reporting on a very worthwhile study of parent-adolescent relationships in the Somali diaspora community in Minnesota. The paper is eminently suitable for publication in this journal and will be of great interest to researchers across disciplines, including intercultural communication, refugee studies and healthcare.

Introduction: The study is situated appropriately in the wider body of research in the subject area, and sufficient theoretical grounding is provided. A little more context could be given on why Minnesota has a relatively large Somali diaspora population. 

Methods: A slightly clearer rationale could be provided as to why the authors chose KI interviews over speaking directly to Somali parents and/or adolescents. I realise that there may well have been access/ethical issues here, but this could at least be acknowledged. The author do mention this as a limitation but a rationale in the methods section would be beneficial. Also, why were individual interviews chosen over focus groups? I can see why one-to-one interview may have been more appropriate here given the subject matter, but the authors could briefly clarify this.  It is not entirely clear why this study was deemed exempt from ethical review (line 151, p. 4). The authors could explain the reason for this. 

Data analysis: Whilst I feel that data analysis is adequately described here, perhaps a little more could be said on how a 'theme' was deemed important.

The findings are presented nicely and the discussion and conclusion is insightful and clearly linked to the data and other relevant literature. Overall, a very enjoyable read which I'm sure will be a great interest to the readership of the journal. 

Author Response

Reviewer 1:

  1. Introduction: The study is situated appropriately in the wider body of research in the subject area, and sufficient theoretical grounding is provided. A little more context could be given on why Minnesota has a relatively large Somali diaspora population. 

Response: Thank you for this suggestion. We have added a sentence to explain factors influencing the Somali diaspora’s resettlement in Minnesota in the paragraph describing the diaspora on page 2.

  1. Methods: A slightly clearer rationale could be provided as to why the authors chose KI interviews over speaking directly to Somali parents and/or adolescents. I realise that there may well have been access/ethical issues here, but this could at least be acknowledged. The author do mention this as a limitation but a rationale in the methods section would be beneficial.

Response: We elected to use key informant (KI) interviews for this formative study to provide a wide range of perspectives from individuals who work with Somali adolescents and their parents that would then guide future focus group research within the diaspora. Recruitment for a larger series of focus groups with adolescents and parents was not feasible within the time or budget constraints for this research project. We have added further information in our methods section on page 4 to describe our rationale for KI interviews.

  1. Also, why were individual interviews chosen over focus groups? I can see why one-to-one interview may have been more appropriate here given the subject matter, but the authors could briefly clarify this. 

Response: We conducted individual interviews to examine the breadth of experiences of KIs in a variety of clinical, public health, educational, religious, and other community organizational settings. Given the diversity of representation of our KIs, individual interviews allowed us to explore perspectives of these professionals in a more free-flowing manner. This approach enabled us to pursue new areas of discussion that may not have emerged in focus groups. We added additional detail in our methods section on page 4 to support our rationale for KI interviews.

  1. It is not entirely clear why this study was deemed exempt from ethical review (line 151, p. 4). The authors could explain the reason for this.

Response: Thank you for pointing out this need for clarification. This study was conducted with KIs who were asked to reflect on their professional experiences serving Somali parents and their adolescent children rather than on their personal experiences. As such, the research was not considered by our institutional review board to be human subject research and it was exempt from review. We have added more detail to our comment on the exemption to reflect this context on page 4.

  1. Data analysis: Whilst I feel that data analysis is adequately described here, perhaps a little more could be said on how a 'theme' was deemed important.

Response: Thank you for this comment. In many ways, we consider all of our emergent themes important. However, through the process of analysis, during which our team constructed meaning though review and consensus, some themes emerged as more prominent or as overarching main themes with associated sub-themes. This was particularly the case where a theme was instrumental in constructing the overall narrative that emerged in the analysis. We have added text to the analysis section on page 5 to provide further clarity. 

  1. The findings are presented nicely and the discussion and conclusion is insightful and clearly linked to the data and other relevant literature. Overall, a very enjoyable read which I'm sure will be a great interest to the readership of the journal.

Response: Thank you. We appreciate your constructive feedback and interest in our research.

Reviewer 2 Report

This is a well-written, interesting paper on parent-child relationships among Somali immigrants in the USA, and how Somali parents navigate the new American (partial) identity of their adolescent children, also in the context of tobacco use. The authors do an excellent job of describing the challenges of both parents and adolescents in navigating acculturation and the experiences of discrimination.

I like the manuscript and its focus on parenting challenges, but I had trouble understanding the focus on tobacco use. The authors argue that the adoption of health-risk behaviors is affected by multi-level effects of immigration (family, peers, community, etc.) - but it does not become clear from the introduction why they consequently focus on smoking/tobacco use. Does tobacco have a specific meaning to Somali individuals? This seems to be suggested by the authors in the discussion (page 11, last paragraph, reference #33) but does not appear anywhere in the text.

In my opinion, the manuscript would tell a complete and interesting story if adolescent tobacco use (and how to support parents in providing guidance to their children) would be left out altogether. Alternatively, the authors need to present a stronger case throughout the manuscript why this is an essential health behavior among this population.

Another issue that needs clarification is the usage of Key Informants (KI) - who were these individuals, why were they chosen (why were they key informants and in relation to what topic, what is their role in parenting and immigration challenges of Somali individuals?), what is their relationship to the parents and children that were the focus of the interviews? I was quite surprised that KI were the respondents rather than the parents and/or the adolescents that were the prime focus of the interviews. The authors should make it clearer why they chose to utilize KI rather than directly interview parents/adolescents.

In general, methods are very concise - too concise to understand completely what the authors did. Especially the section on recruitment of participants (inclusion/exclusion criteria), interview guide, and data analysis (how many themes?) need more details. It is not sufficient to refer to another article in which these details are provided. 

Author Response

Reviewer 2:

  1. The authors do an excellent job of describing the challenges of both parents and adolescents in navigating acculturation and the experiences of discrimination. I like the manuscript and its focus on parenting challenges, but I had trouble understanding the focus on tobacco use. The authors argue that the adoption of health-risk behaviors is affected by multi-level effects of immigration (family, peers, community, etc.) - but it does not become clear from the introduction why they consequently focus on smoking/tobacco use. Does tobacco have a specific meaning to Somali individuals? This seems to be suggested by the authors in the discussion (page 11, last paragraph, reference #33) but does not appear anywhere in the text. In my opinion, the manuscript would tell a complete and interesting story if adolescent tobacco use (and how to support parents in providing guidance to their children) would be left out altogether. Alternatively, the authors need to present a stronger case throughout the manuscript why this is an essential health behavior among this population.

Response: Thank you for raising this concern. Tobacco use is a particularly challenging risk behavior facing Somali adolescents given the high levels of adult tobacco use (both combustible cigarettes and water pipe use) and the prevalence of tobacco product use among Somali adolescents in Minnesota. We moved our description of these challenges and the rationale for focusing on Somali adolescent tobacco prevention to page 2 just after introducing the Somali diaspora. We also added further rationale for tobacco use as an increasingly prevalent risk behavior in our framing of the research questions on page 3.

  1. Another issue that needs clarification is the usage of Key Informants (KI) - who were these individuals, why were they chosen (why were they key informants and in relation to what topic, what is their role in parenting and immigration challenges of Somali individuals?), what is their relationship to the parents and children that were the focus of the interviews? I was quite surprised that KI were the respondents rather than the parents and/or the adolescents that were the prime focus of the interviews. The authors should make it clearer why they chose to utilize KI rather than directly interview parents/adolescents.

Response: We elected to use key informant (KI) interviews for this formative study to provide a wide range of perspectives from individuals who work in a professional capacity with Somali adolescents and their parents to guide future focus group research within the diaspora. KIs are individuals who current work or have worked recently with Somali adolescents and their families in their professional roles as doctors, community health workers, imams, teachers, and in community organization roles that provide outreach and support within the Somali diaspora. Recruitment for a larger series of focus groups with adolescents and parents was not feasible within the time and budget constraints for this formative research project. We have added further information in our methods section on page 4 to describe our rationale for KI interviews and more details about the professional roles that the KIs represented.

  1. In general, methods are very concise - too concise to understand completely what the authors did. Especially the section on recruitment of participants (inclusion/exclusion criteria), interview guide, and data analysis (how many themes?) need more details. It is not sufficient to refer to another article in which these details are provided. 

Response: Thank you for the suggestion to add in more detail to our methods section. We have added in more information on the inclusion and exclusion criteria for this study and examples of the professional roles of our KIs within our methods section on page 4. We have also included a copy of the interview guide that reflects the full scope of our interviews as Appendix 1 and have referenced this when discussing the content of the interviews. In response to Reviewer 1 suggestions, we have also included additional detail in the data analysis section.

Reviewer 3 Report

Referee:

The paper  “Belonging to three worlds: Somali adolescent-parent relationships in the United States and implications for tobacco prevention” introduces very interesting issues. However major revisions are needed for its publication.

 The aim of the study was to examine qualitatively how Somali adolescent-parent relationship factors influence Somali adolescent tobacco use and to identify areas for further family support to prevent Somali adolescent tobacco use.

To reach the aim of the study the authors conducted fifteen key informant interviews with professionals serving the Somali community in clinical, educational, religious, or other community organization roles in one Minnesota metropolitan region. So the authors examined indirectly the topic of their research. All issues connected with this shift from Somali adolescents and their families and professionals are not clearly clarify.

It is not clear form the paper first how Somali immigrants are involved with community organizations etc. How it works. How professionals are in touch with Somali adolescents, their families and tobacco issues.

All the background about the informant interviews are really not clear.

The interviews were conducted in English. What it means that may be some of the possible participants would be unable to understand English?

The authors really need to clarify “Participants in this study were English-proficient professionals (both Somali and non-Somali) over 18 years of age who served the greater metropolitan Somali community in Minnesota in clinical, public health, educational, religious, and other community organizational settings. The research team used purposive sampling techniques to identify potential KIs who met inclusion criteria for this study, building off known contacts of team members …. In total, 24 experts 149 were contacted, of which fifteen agreed to participate.” Information about informants and their role also is not well identified. From the text it appears that sometimes they are talking as parents, sometime only as professional. This is again an issue.

The biggest problem is about the participants in the study. Because finally participants are mostly Somali and so their interviews are connected with  ftheir roots. This is very important but need to be stressed and clarify. Most of the qualitative data reported are influenced by this very basic issue and needs to be taken into account in presenting the results and in the conclusion of the paper as well as in the limitations.

Some other few comments about the description of the sample.

Table 1 and Table 2. Key informant characteristics need to be summarize in a more structural way. Some of the percentages instead of making easier to describe the sample they make the information more dispersive. I understand that it is important to know the mean number of years working with Somali youth, but with what role? How they were in touch and how with their families? 

Author Response

Reviewer 3: 

  1. The aim of the study was to examine qualitatively how Somali adolescent-parent relationship factors influence Somali adolescent tobacco use and to identify areas for further family support to prevent Somali adolescent tobacco use. To reach the aim of the study the authors conducted fifteen key informant interviews with professionals serving the Somali community in clinical, educational, religious, or other community organization roles in one Minnesota metropolitan region. So the authors examined indirectly the topic of their research. All issues connected with this shift from Somali adolescents and their families and professionals are not clearly clarify.

Response: Thank you for the opportunity to clarify this point. We elected to use key informant (KI) interviews for this formative study to provide a wide range of perspectives from individuals who work with Somali adolescents and their parents that would then guide future focus group research within the diaspora. Recruitment for a larger series of focus groups with adolescents and parents was not feasible with the time and budget constraints for this research project. We have added further information in our methods section on page 4 to describe our rationale for KI interviews and have responded to the Reviewer’s specific questions regarding our methods below.

  1. It is not clear form the paper first how Somali immigrants are involved with community organizations etc. How it works. How professionals are in touch with Somali adolescents, their families and tobacco issues.

Response: We agree that we could provide further clarity on the identities of our KIs and the community organizations that they represent. We have added additional detail regarding the professional roles of our KIs that involved working directly with Somali youth and their families in our methods section on page 4 in response to this Reviewer and the other Reviewers’ suggestions.

  1. All the background about the informant interviews are really not clear. The interviews were conducted in English. What it means that may be some of the possible participants would be unable to understand English?

Response: We conducted KI interviews with professionals who serve the Somali community. All of our participants were college-educated (often with graduate-level degrees) and were bilingual, so language barriers were not an issue. We have clarified this in our inclusion/exclusion criteria in our methods section on page 4. 

  1. The authors really need to clarify “Participants in this study were English-proficient professionals (both Somali and non-Somali)over 18 years of age who served the greater metropolitan Somali community in Minnesota in clinical, public health, educational, religious, and other community organizational settings. The research team used purposive sampling techniques to identify potential KIs who met inclusion criteria for this study, building off known contacts of team members …. In total, 24 experts 149 were contacted, of which fifteen agreed to participate.” Information about informants and their role also is not well identified. From the text it appears that sometimes they are talking as parents, sometime only as professional. This is again an issue.

Response: This study was conducted with KIs who were asked to reflect on their professional experiences serving Somali parents and their adolescent children rather than on their personal experiences. However, some of our Somali-identifying KIs shared aspects of their personal experiences as parents or community members when responding to questions. We have added more detail within our methods section on page 4 to describe the roles that participants played within their organizations and the nature of the questions asked. We also referenced our interview guide on page 4 and provided this guide as Appendix 1.

  1. The biggest problem is about the participants in the study. Because finally participants are mostly Somali and so their interviews are connected with their roots. This is very important but need to be stressed and clarify. Most of the qualitative data reported are influenced by this very basic issue and needs to be taken into account in presenting the results and in the conclusion of the paper as well as in the limitations.

Response: We agree with the Reviewer that the influence of our Somali KIs on our results is very important to their interpretation. In our opinion, the inclusion of a predominantly Somali-identifying group of KIs across a broad age-spectrum is a strength of our study as it provides an opportunity to explore experiences from within the diaspora by speaking with members who have firsthand knowledge of these experiences. We added a sentence providing more context on the perspectives of our Somali KIs to the first paragraph of our discussion section on page 10. We rephrased our limitations sentence on this topic to “the majority of our Somali KIs were Somali parents and young adults” to acknowledge the influence of their life experiences on their perspectives. Finally, we included a phrase in our conclusions section on page 13.

  1. Some other few comments about the description of the sample. Table 1 and Table 2. Key informant characteristics need to be summarize in a more structural way. Some of the percentages instead of making easier to describe the sample they make the information more dispersive. I understand that it is important to know the mean number of years working with Somali youth, but with what role? How they were in touch and how with their families? 

Response: Thank you for your suggestion. Our manuscript contains one table. In Table 1, we initially provided a high-level overview of our KIs demographics including the organization in which they work professionally and the length of time that they have been working with Somali youth. We agree with the reviewer that we could provide more detailed information on the mean number of years working with Somali youth broken out by community organization type. We have made these changes to Table 1 on page 5 to reflect this detail. We have included more information on the community organizations and some of the specific roles that our KIs represented in our methods section on page 4.

Round 2

Reviewer 2 Report

Thank you for your revisions and the responses to my comments. I think the rationale for the study and the use of the sample are now much clearer, and I have no further comments. The manuscript is very well written and interesting! 

Author Response

Reviewer 2:

  1. Thank you for your revisions and the responses to my comments. I think the rationale for the study and the use of the sample are now much clearer, and I have no further comments. The manuscript is very well written and interesting! 

Response: Thank you for your help in strengthening our manuscript.

Reviewer 3 Report

I think the authors have answered to my comments. However, I have seen the Interview is only published in the supplementary data. I think it would be important to insert it in the paper as Appendix 1. 

Two other small comments. First I do not know if everyone knows what is hooka. Second I do not think that in the description of the sample authors has to specify that participants are older than 18 years of age. Of course, they are adults. 

Author Response

  1. I think the authors have answered to my comments. However, I have seen the Interview Guide is only published in the supplementary data. I think it would be important to insert it in the paper as Appendix 1.

Response: We agree with the Reviewer that the best placement of this information is likely as an Appendix. We refer to our interview guide supplement as Appendix 1 in the text and have attached the guide in the submitted documents. We leave it up to the editors to determine the best placement of the guide within the publication.

  1. Two other small comments. First I do not know if everyone knows what is hookah.

Response: Thank you for pointing out this opportunity to clarify the definition of hookah. We have added in a definition of hookah as waterpipes at the first mention of the term on page 2.

  1. Second I do not think that in the description of the sample authors has to specify that participants are older than 18 years of age. Of course, they are adults. 

Response: We changed the inclusion criteria to read as “adults” instead of 18 years or older.